# Fertility Preservation in Early-Stage Endometrial Carcinoma and EIN: A Single-Centre Experience and Literature Review

**DOI:** 10.3390/cancers17213464

**Published:** 2025-10-28

**Authors:** Zoárd Tibor Krasznai, Emese Hajagos, Vera Gabriella Kiss, Péter Damjanovich, Sára Tóth, Szabolcs Molnár

**Affiliations:** Department of Obstetrics and Gynaecology, University of Debrecen, Nagyerdei krt. 98, 4032 Debrecen, Hungary; krasznai.zoard@med.unideb.hu (Z.T.K.); hajagos.emese@med.unideb.hu (E.H.); dr.kiss.gabriella@med.unideb.hu (V.G.K.); damjanovich.peter@med.unideb.hu (P.D.); toth.sara100@gmail.com (S.T.)

**Keywords:** early-stage endometrial cancer, endometrial intraepithelial neoplasia, fertility-sparing therapy

## Abstract

**Simple Summary:**

Endometrial cancer is the most common gynaecological malignancy in developed countries, with rising incidence and mortality. In reproductive-age women diagnosed with early-stage disease or endometrial intraepithelial neoplasia, fertility-sparing treatment may be a viable option. This retrospective study evaluated 13 patients treated at the University of Debrecen between 2018 and 2023. The majority of patients responded well to therapy, and several pregnancies were able to be achieved. These findings support the feasibility of conservative management in selected cases, emphasising the importance of a multidisciplinary approach.

**Abstract:**

**Objectives:** Endometrial carcinoma is the most common gynaecological cancer in developed countries, with both incidence and mortality rates continuing to rise globally. For women of reproductive age diagnosed with early-stage disease or endometrial intraepithelial neoplasia, fertility-preserving treatment should be considered to maintain the possibility of future childbearing. Effective fertility-sparing management requires a multidisciplinary approach that includes patient education, reduction in risk factors, accurate molecular and histological classification to guide targeted therapies, assisted reproductive technologies to improve early conception rates, and attention to the psycho-sexual well-being of patients to support treatment adherence. **Methods:** This retrospective cohort study analysed the clinicopathological features and treatment outcomes of thirteen patients who received fertility-preserving therapy between 2018 and 2023. **Results:** The mean age of the patients (*n* = 13) was 34.4 years, with a range of 20 to 41 years. The overall treatment response rate was 76.9%, including 69.2% complete and 7.7% partial responses. Stable disease was observed in 15.4% of cases, while progression occurred in 7.7%. Among those who achieved complete remission, in vitro fertilisation (IVF) was initiated in four cases, with two ongoing as of the time of data analysis. In one of the cases, after two unsuccessful assisted reproductive attempts, spontaneous conception occurred, resulting in the birth of a child. **Conclusions:** Our findings support the feasibility and success of fertility-preserving treatment in carefully selected patients, allowing the preservation of reproductive potential alongside oncological care.

## 1. Introduction

Endometrial carcinoma (EC) is the most common gynaecologic malignancy in high-income countries, with its incidence steadily increasing due to demographic and lifestyle factors such as population aging, obesity, insulin resistance, and unopposed estrogen exposure [1,2]. Globally, the disease burden is rising, with an estimated 417,000 new cases and 97,000 deaths reported in 2020, making it a growing public health concern [3]. While EC typically affects postmenopausal women, approximately 4–14% of cases occur in women under the age of 40, many of whom have not yet completed childbearing or have never conceived [4]. In Hungary, data from the National Cancer Registry show that in 2000, 41 out of 1480 patients diagnosed with uterine malignancy (2.77%) were under 40, increasing to 56 out of 1961 patients (2.86%) by 2022 [5].

The precursor lesion, endometrial intraepithelial neoplasia (EIN), carries a 30–50% risk of progression to carcinoma if left untreated [6]. In reproductive-age women diagnosed with EIN or well-differentiated, early-stage (FIGO IA, grade 1 endometrioid) endometrial cancer confined to the endometrium, fertility preservation is an increasingly relevant therapeutic goal. Standard treatment for EC—total hysterectomy with bilateral salpingo-oophorectomy—eliminates reproductive potential, making it a less desirable option for patients desiring future fertility [7,8].

Fertility-sparing treatment in this context typically involves the use of high-dose progestin therapy, either orally (e.g., medroxyprogesterone acetate, megestrol acetate) or via levonorgestrel-releasing intrauterine devices (LNG-IUD), sometimes in combination with hysteroscopic tumour resection [9,10]. Response rates vary widely across studies, with complete response achieved in approximately 76–86% of cases; however, recurrence is not uncommon and necessitates strict follow-up [9]. Assisted reproductive technologies, such as in vitro fertilisation (IVF), are often necessary to optimise the likelihood of pregnancy once remission is achieved [11,12]. Despite these efforts, pregnancy and live birth rates remain variable, underscoring the complexity of balancing oncologic safety with reproductive success [9,13].

The advent of molecular classification in EC, particularly the 2013 TCGA (The Cancer Genome Atlas) framework, has added an essential dimension to patient selection and risk stratification [14,15]. Identifying molecular subgroups, such as POLE ultramutated, mismatch repair-deficient (dMMR), p53-abnormal, and those with no specific molecular profile (NSMP), can refine treatment decisions and predict outcomes, although the role of these classifications in fertility-sparing therapy remains under investigation [16,17].

Given the rarity and complexity of these cases, real-world data from single-centre experiences provide valuable insights that can inform clinical guidelines and support evidence-based decision-making. In this study, we present a retrospective analysis of fertility-preserving management in patients with early-stage endometrial carcinoma and EIN treated at a tertiary referral centre between 2018 and 2023. We also provide a comprehensive review of the current literature, aiming to contextualise our findings within the broader landscape of fertility preservation in endometrial neoplasia.

## 2. Materials and Methods

### 2.1. Study Design and Patient Selection

This retrospective cohort study included thirteen women under the age of 45 diagnosed with endometrial intraepithelial neoplasia (EIN) or early-stage endometrioid endometrial carcinoma (FIGO stage IA, grade 1 or 2) between 2018 and 2023. All patients were treated at the Department of Obstetrics and Gynaecology, University of Debrecen, Hungary. The study was approved by the Regional and Institutional Research Ethics Committee (RKEB/IKEB 6562-2023). All patients received detailed counseling about the non-standard nature of fertility-sparing therapy, including its risks and potential benefits. A multidisciplinary tumour board approved treatment initiation in each case.

### 2.2. Diagnostic Workup

All patients underwent a gynaecological examination followed by a transvaginal ultrasound (TVUS) to assess endometrial thickness, and a histological sample was obtained via endometrial biopsy or diagnostic hysteroscopy. Following histological confirmation of EIN or carcinoma, imaging studies were performed for staging, including pelvic MRI to assess myometrial and cervical involvement, and chest and abdominal CT to confirm that locally the tumour is limited to the endometrium, and to exclude distant metastases.

### 2.3. Fertility-Sparing Treatment

Before the initiation of hormonal treatment, the eligible patients underwent hysteroscopic resection of visible lesions. Hormonal treatment included oral medroxyprogesterone acetate (MPA, 400–600 mg/day) and/or placement of a levonorgestrel-releasing intrauterine system (LNG-IUS; 52 mg). The patients continued the hormone therapy up to 12 months; the maximum treatment duration was 15 months in cases where remission was not achieved earlier.

### 2.4. Follow-Up and Response Assessment

Treatment response was monitored via TVUS and pelvic examination every 3 months. Hysteroscopy with endometrial sampling was performed every 3 to 6 months. Complete remission was defined as two consecutive negative histological samples taken at least 3 months apart. Surgical management was recommended if no response was observed at 6 months or progression occurred at any time.

Patients achieving complete remission continued maintenance hormonal therapy until family planning. Assisted reproductive technologies (ART) were recommended to minimise the time to pregnancy and reduce the risk of recurrence. Hysterectomy with or without salpingo-oophorectomy was performed in cases of disease progression, recurrence, or when childbearing was no longer desired.

### 2.5. Statistical Analysis

Data were analysed using descriptive statistics. Categorical variables were summarised as frequencies and percentages, while continuous variables were reported as means with ranges. No inferential statistical testing was performed due to the small sample size.

## 3. Results

### 3.1. Patient Characteristics

Thirteen patients were included in the study. The mean age of patients was 34.38 years, and the average body mass index (BMI) was 35.85 (±12.48 kg/m^2^). The most common comorbidities were obesity (69%), polycystic ovary syndrome (46%), hypothyroidism (46%), and insulin resistance (31%). Four patients had a history of pregnancy, and two had previously given birth. Histologically, we diagnosed 11 patients (81.8%) with endometrial carcinoma (EC) and 2 (18.2%) with endometrial intraepithelial neoplasia (EIN). Among the EC cases, 6 were FIGO stage IA grade 1, and 5 were FIGO stage IA grade 2 (Table 1).

### 3.2. Oncological Outcomes

The overall response rate to fertility-sparing therapy was 76.9%. Complete response (CR), defined as two consecutive negative histological samples, was achieved in 9 patients (69.2%). One patient (7.7%) experienced a partial response (PR), indicated by histological regression from EC to EIN. Stable disease (SD), characterised by the absence of histological regression, was observed in 2 cases (15.4%), and progressive disease (PD), defined by advancement in tumour grade or stage, was noted in 1 case (7.7%) In terms of outcomes we must pay special attention to patients with G2 disease, since 2 of 5 (40%) had no response to treatment, 1 (20%) had progression under treatment based on final histology, and two patients are still in the follow-up phase. Comparing these results with the G1 subgroup, in which none of the six patients were non-responders to treatment, the G2 histology was an unfavourable prognostic factor (Table 2).

Thus, fertility-sparing treatment was successful in 9 out of 13 patients (69.2%), who achieved complete remission, as confirmed by follow-up histological evaluation.

### 3.3. Treatment Failures and Surgical Outcomes

The average treatment duration of the four patients with unsuccessful outcomes was 170 days. Patient I, initially diagnosed with FIGO IA grade 1 EC, achieved partial remission (regression to EIN); however, due to persistent histological abnormalities, total laparoscopic hysterectomy (TLH) was performed. The final histology revealed endometrial hyperplasia without atypia. Patients II and III with FIGO IA grade 2 EC demonstrated stable disease on follow-up biopsies. TLH and total abdominal hysterectomy (TAH) were performed, respectively. The final histological evaluation confirmed the presence of persistent or recurrent adenocarcinoma. Patient IV, also initially diagnosed with FIGO IA grade 2 EC, showed disease progression on follow-up biopsy, prompting TAH. Postoperative histopathology revealed an upgrade to FIGO stage IIIA adenocarcinoma (Table 3).

These findings underscore the importance of cautious patient selection in grade 2 cases, where a higher risk of treatment failure and progression appears to be present.

### 3.4. Reproductive Outcomes

Among the nine patients who achieved complete remission, four initiated in vitro fertilisation (IVF) treatment, with two cases ongoing at the time of analysis. In one patient, following two unsuccessful ART cycles, spontaneous conception occurred, resulting in the birth of a healthy child. One patient is expecting her baby following successful ART. The fertility rate among the population intending to conceive was 50%. Among grade 2 cases, one patient underwent ART, with no success at the time of article submission. 

## 4. Discussion

This retrospective analysis of thirteen patients treated with fertility-preserving therapy for endometrial intraepithelial neoplasia (EIN) or early-stage endometrioid endometrial carcinoma (EC) highlights the potential success of conservative management in carefully selected young women. With a complete response rate of 69.2% and an overall response rate of 76.9%, our findings align with those of previously published studies and meta-analyses, confirming the viability of this approach in clinical practice. Our study population reflects standard clinical features of patients eligible for fertility preservation, including a relatively young mean age (34.4 years), high prevalence of obesity and polycystic ovary syndrome (PCOS), and early-stage, low-grade endometrioid histology. These characteristics are consistent with the well-established risk profile for type I endometrial cancer, which is closely linked to estrogen-driven pathogenesis and metabolic factors such as insulin resistance and adiposity [18,19]. One of the key observations in our cohort was the relatively high success rate of hormonal treatment when combined with hysteroscopic tumour resection and close follow-up. Similar combined approaches have shown favourable outcomes in international studies, often improving the rate and speed of histological remission compared to hormonal therapy alone [20,21]. The use of medroxyprogesterone acetate (MPA) or levonorgestrel-releasing intrauterine systems (LNG-IUS), both of which are used in our patient group, is well supported by existing guidelines and literature [22]. Importantly, our results also reinforce the caution needed when extending fertility-sparing treatment to patients with grade 2 tumours. Among the four patients in our cohort who failed to achieve remission or experienced disease progression, three had FIGO grade 2 histology at baseline. One of these patients progressed to stage IIIA carcinoma during treatment, underscoring the oncologic risk associated with attempting conservative management outside of current ESGO/ESHRE/ESGE guideline indications [23]. These findings support a more conservative, individualised approach to managing higher-grade lesions, ideally incorporating molecular profiling into the risk-stratification process.

Although pregnancy rates remain modest, the fact that one patient conceived spontaneously and 4 initiated IVF following complete remission reflects the real-world feasibility of post-treatment reproduction. However, our cohort also illustrates the challenges in achieving pregnancy, likely due to confounding factors such as obesity, delayed ART referral, and underlying subfertility. Studies have shown that live birth rates following fertility-sparing therapy can range from 20% to 30%, with significantly better outcomes when ART is initiated promptly after achieving remission [9,24]. In our cases, IVF cycles started after the removal of levonorgestrel IUD and/or the stop of hormonal treatment. However, there is growing evidence that stimulation with letrozole is safe with the levonorgestrel intrauterine system (LNG-IUS) in situ. Thus, oocyte retrieval is possible with the IUS still in situ, and embryo transfer(s) can be performed after IUS removal, further increasing fertility rates [25].

Fertility preservation in women with early-stage endometrial carcinoma (EC) and endometrial intraepithelial neoplasia (EIN) has become an increasingly relevant clinical goal due to the rising incidence of EC among reproductive-aged women. Although the standard treatment for EC involves total hysterectomy with bilateral salpingo-oophorectomy, which irreversibly eliminates fertility, conservative management can be considered in selected cases. The growing body of evidence from retrospective cohorts, meta-analyses, and guideline recommendations supports the feasibility of hormonal treatment for fertility preservation, provided strict selection criteria are applied and multidisciplinary follow-up is ensured [19,23].

Current international guidelines, including those from ESGO/ESHRE/ESGE, recommend fertility-sparing treatment only in highly selected patients. This includes women with well-differentiated (grade 1), endometrioid-type EC limited to the endometrium (FIGO stage IA), without evidence of myometrial invasion, lymphovascular space invasion (LVSI), or extrauterine spread. Molecular profiling, where available, is also encouraged to exclude high-risk histological or genomic subtypes [23,26]. For women with EIN, which is considered a precancerous lesion with a 30–50% progression risk, fertility-sparing management is deemed safe and effective in the absence of atypical or aggressive features [6,27].

Regarding oncological outcomes, numerous studies have confirmed the efficacy of progestin-based therapy. A comprehensive meta-analysis by Gallos et al. (2012) included 34 studies and over 550 patients, reporting a regression rate of 76.2% in EC population, with a relapse rate of 40.6% and a live birth rate of 28% and a 85.6% regression rate in EIN population, with a relapse rate of 26% and a live birth rate of 26.3 [9]. Comparing our results, a wide range of treatments was applied in the studies; hysteroscopic tumour resection was reported in only one study, where the response rate was the highest [9,28]. Our results are also comparable to those of Park et al. (2013), who examined 48 women treated with oral progestins and reported a CR rate of 77.1%, recurrence in 30.5%, and disease progression in 5.5% [10]. Recurrence rates varied between subgroups stratified by tumour grade, with rates of 23.1%, 47.1%, and 71.4% in grades 1, 2, and 3, respectively [10]. Similarly, Koskas et al. (2014) analysed a multicenter cohort of 74 patients and found a CR rate of 81.1% and a recurrence rate of 20.3% [29].

The inclusion of patients with grade 2 EC in our cohort, although limited, is noteworthy. While most guidelines restrict fertility preservation to grade 1 disease, several studies have explored its use in grade 2 EC with mixed results. Falcone et al. (2017) reported a CR rate of 65.2% in 23 women with grade 2 EC, but also noted higher recurrence and progression rates than in patients with grade 1 EC [30]. In our study, three out of four patients who failed treatment had grade 2 disease, and one progressed to stage IIIA carcinoma, suggesting that while grade 2 EC may respond to hormonal therapy in some cases, it carries a significantly higher oncologic risk.

Reproductive outcomes following fertility-sparing treatment remain variable. While achieving remission is often feasible, the ability to conceive and deliver a healthy child depends on multiple factors, including patient age, baseline fertility, BMI, access to assisted reproductive technologies (ARTs) and the patient’s complex life situation after the operation, regarding her later willingness to conceive. In our cohort, four patients initiated in vitro fertilisation (IVF) after complete remission, and one woman achieved a spontaneous pregnancy resulting in a live birth, and one has had a successful conception recently. These figures are consistent with the broader literature. Gallos et al. (2012) reported a live birth rate of 28% (21.6–36.3, 95% CI) [9]. Koskas et al. (2014) highlighted that early referral for reproductive care improves time to conception and overall pregnancy success [29].

Overall, the findings from our study are consistent with the existing literature, supporting the use of fertility-sparing therapy in highly selected cases of EIN and early-stage EC. Treatment success is closely tied to the selection of appropriate patients, individualised treatment planning, and a timely transition to definitive surgery when necessary. Reproductive outcomes can be optimised through early ART involvement and management of comorbidities. Future research should focus on prospective registries, integration of molecular classification, and development of novel combined hormonal or immunological treatments to improve remission durability and reduce relapse.

Our study has several limitations, including the small sample size, retrospective design, and limited availability of molecular data. Nonetheless, the real-world experience presented here contributes to the growing body of evidence supporting fertility preservation in early-stage EC and EIN, while also highlighting the need for cautious patient selection, multidisciplinary care, and long-term oncologic surveillance.

## 5. Conclusions

Fertility-sparing therapy represents a feasible and effective option for carefully selected young women with endometrial intraepithelial neoplasia (EIN) or early-stage, low-grade endometrioid endometrial carcinoma (EC). Our results, showing a complete remission rate of 69.2%, are consistent with international data and reinforce the importance of combined approaches—such as hysteroscopic resection followed by progestin therapy—supported by close multidisciplinary follow-up. While pregnancy outcomes remain modest, early referral for assisted reproduction and optimisation of metabolic comorbidities may improve success rates. Given the higher oncologic risk in grade 2 EC, treatment should remain individualised and confined to experienced centres. Future studies incorporating molecular profiling and prospective data collection are warranted to refine patient selection and improve long-term outcomes.

## Figures and Tables

**Table 1 cancers-17-03464-t001:** Characteristics of the patient population used in the research.

**Number of patients**	13 (100%)
**Average age at diagnosis**	34.38
**BMI (body mass index, kg/m^2^)**	35.85 ± 12.48
**Most common medical history**	
Obesity	9 (69%)
PCOS	6 (46%)
Hypothyroidism	6 (46%)
Insulin resistance	4 (31%)
**Obstetric history**	
Previous pregnancies	4
Previous live birth	2
**Type of histology**	
EIN	2 (18.2%)
EC	11 (81.8%)
FIGO Stage IA G1	6
FIGO Stage IA G2	5

**Table 2 cancers-17-03464-t002:** Detailed clinical characteristics of study participants.

Pt. #	Age at Diagnosis	BMI	Medical History	Previous Livebirth	Histology	Grade	FIGO Stage	Time to Response (Months)	Best Response	Definitive Treatment	Final Histology
#1	20	60.8	Obesity, PCOS, Hypothyroidism, IR	No	EIN	NA	NA	12	CR		
#2	29	31.2	Obesity, PCOS	Yes	EC	G1	IA	-	PR	TLH	EH
#3	27	30.8	Obesity, PCOS, Hypothyroidism, Hyperprolactinemia, Hypertension	No	EC	G2	IA	-	SD	TLH	EC FIGO IA
#4	32	36.3	PCOS, Obesity, IR, Hypertension	No	EC	G2	IA	4	CR		
#5	34	49.0	Obesity, PCOS, IR, Hypothyroidism	No	EC	G1	IA	3	CR		
#6	35	40.0	Obesity, PCOS	No	EC	G2	IA	-	SD	TAH	EC FIGO IA
#7	34	29.1	-	No	EC	G2	IA	15	CR		
#8	36	51.1	Obesity, Hypothyroidism	No	EC	G2	IA	-	PD	TAH	EC FIGO IIIA
#9	38	43.5	Obesity, Hypothyroidism, IR, Hyperprolactinemia	No	EC	G1	IA	10	CR		
#10	35	19.5	-	No	EIN	NA	NA	14	CR		
#11	42	21.8	Chronic anemia	Yes	EC	G1	IA	4	CR		
#12	42	30.3	Hypertension, Hypothyroidism, Depression, Obesity	No	EC	G1	IA	5	CR		
#13	42	22.6	-	No	EC	G1	IA	7	CR		
CR (complete response)				9 (69.20%)							
PR (partial response)				1 (7.70%)							
SD (stable disease)				2 (15.40%)							
PD (progressive disease)				1 (7.70%)							

**Table 3 cancers-17-03464-t003:** A detailed description of failed cases.

	Patient I.	Patient II.	Patient III.	Patient IV.
**Histology type**	EC	EC	EC	EC
**Grade**	G1	G2	G2	G2
**FIGO STAGE (2018)**	IA	IA	IA	IA
**First control sampling**	HSC, EIN	HSC, EIN	HSC, EC	HSC, EC
**Second control sampling**	HSC, EIN	HSC, EC	-	-
**Best response to treatment**	PR	SD	SD	PD
**Hysterectomy**	TLH	TLH	TAH	TAH
**Histopathological result**	endometrial hyperplasia without atypia	adenocarcinoma endometrii, Stage IA	endometrioid adenocarcinoma, Stage IA	adenocarcinoma endometrioides endometrii, Stage IIIA

Average length of fertility preservation treatment: 170 days. Abbreviations: EC: endometrial carcinoma, G: grade, HSC: hysteroscopy, EIN: endometrial intraepithelial neoplasia, PR: partial response, SD: stable disease, PD: progressive disease, TLH: total laparoscopic hysterectomy, TAH: total abdominal hysterectomy.

## Data Availability

The authors declare that the data of this research is available from the corresponding author on request.

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
