# Peer review of "Fertility Preservation in Early-Stage Endometrial Carcinoma and EIN: A Single-Centre Experience and Literature Review"

_cancers, 2025, doi:10.3390/cancers17213464_

Round 1

Reviewer 1 Report

Comments and Suggestions for Authors

I enjoyed reading this paper because it is honest, both in its writing and in its criticisms, as the authors were able to sincerely highlight the text's shortcomings. However, I think it provides a detailed and observational snapshot of both the problem and the possible intervention, leaving room for further investigation. I hope the authors can also follow up on the patients they follow, both in terms of recurrence (with their full analysis) and in terms of any pregnancies.

Two clarifications:

1. To make a rationale, I cannot understand which patients failed treatment by trying to correlate Table 3 with Table 2. Please indicate this.

2. Specifically regarding the discussion about G2 patients being the majority of patients who failed treatment, could you please discuss what happened to the other G2 patients and whether there is a relationship between stage, improved treatment outcome, and fertilization? Perhaps with a table or graph? In this way the reader better understands a point that is important to me and which would also include a discussion on prevention.

Author Response

Dear Reviewer,

We appreciate the kind and constructive comments. We greatly appreciate the encouraging words regarding the honesty and balance of our manuscript. We believe the constructive remarks have significantly improved the scientific quality of the paper. Below, please find the detailed answer and corrections to your questions and comments.

1. To make a rationale, I cannot understand which patients failed treatment by trying to correlate Table 3 with Table 2. Please indicate this.

Answer:

We have revised Table 2 to clearly indicate which patients experienced treatment failure, making the correlation between Table 2 and Table 3straightforward and easily traceable. The patients’ previous history and relevant characteristics are now also included for better clarity (lines 149-153 and table 2).

2. Specifically regarding the discussion about G2 patients being the majority of patients who failed treatment, could you please discuss what happened to the other G2 patients and whether there is a relationship between stage, improved treatment outcome, and fertilization? Perhaps with a table or graph? In this way the reader better understands a point that is important to me and which would also include a discussion on prevention.

Answer:

We have expanded the section on reproductive outcomes to include additional information about grade 2 cases (Lines 149-154), and we have included it in the conclusion as well (Lines 287-299). Specifically, we now note that among the two G2 patients who achieved complete remission, one underwent ART, which has been unsuccessful so far (Lines 179-181).

We believe these additions improve the understanding of treatment response and reproductive outcomes in this subgroup, in line with the reviewer’s valuable suggestion.

Reviewer 2 Report

Comments and Suggestions for Authors

This is a retrospective study analyzed the clinicopathological features and treatment outcomes of 13 patients with EIN or early-stage endometrioid endometrial carcinoma who received fertility-preserving therapy between 2018 and 2023.

This is an interesting topic for the readers despite the small number of participants.

Patients caracteristics are presented adequately. Please consider to present all the clinical aspects of the patients in comparison with the histological findings in a new table in order to give all the information toghether.

Limitations are the small sample size and the absence of molecular info of the tumors, which are stated clearly as limtations.

Author Response

Dear Reviewer,

We sincerely thank for the positive and encouraging evaluation of our work. We appreciate the thoughtful suggestion regarding the presentation of clinical and histological data. We believe your suggestions have contributed to improving our paper.

Following the reviewer’s recommendation, we have substantially revised Table 2, which now clearly summarizes all relevant clinical characteristics of the patients alongside the corresponding histopathological findings. This modification, in line with the reviewer’s valuable advice, allows for a more comprehensive and transparent presentation of the data.